# Assessment of the Genetic Diversity and Population Structure of *Rhizophora mucronata* along Coastal Areas in Thailand

**DOI:** 10.3390/biology12030484

**Published:** 2023-03-21

**Authors:** Chaiwat Naktang, Supaporn Khanbo, Chutintorn Yundaeng, Sonicha U-thoomporn, Wasitthee Kongkachana, Darunee Jiumjamrassil, Chatree Maknual, Poonsri Wanthongchai, Sithichoke Tangphatsornruang, Wirulda Pootakham

**Affiliations:** 1National Omics Center, National Science and Technology Development Agency, 113 Thailand Science Park, Khlong Luang, Pathum Thani 12120, Thailand; 2Department of Marine and Coastal Resources, 120 The Government Complex, Chaengwatthana Rd., Thung Song Hong, Bangkok 10210, Thailand

**Keywords:** mangrove, *Rhizophora mucronata*, Rhizophoraceae, whole-genome, genetic diversity, population structure, SNP

## Abstract

**Simple Summary:**

In order to examine the genetic diversity and population structure of the *Rhizophora mucronata* population in Thailand, we utilized 10× Genomics technology to obtain a comprehensive whole-genome sequence, and restriction site associated DNA sequencing (RAD-seq) to genotype the population. Using SNPs discovered from the *R. mucronata* genome sequence, we detected moderate levels of genetic diversity and differentiation across 80 *R. mucronata* accessions collected from the coastal regions of Thailand. Both population structure and principal component analysis (PCA) converged on a clustering of two subpopulations. However, the results of two genetic groups did not correspond to the Gulf of Thailand or the Andaman Sea. Several factors could have influenced the *R. mucronata* genetic pattern, such as hybridization and anthropogenic factors.

**Abstract:**

Unique and biodiverse, mangrove ecosystems provide humans with benefits and contribute to coastal protection. *Rhizophora mucronata*, a member of the Rhizophoraceae family, is prevalent in the mangrove forests of Thailand. *R. mucronata*’s population structure and genetic diversity have received scant attention. Here, we sequenced the entire genome of *R. mucronata* using 10× Genomics technology and obtained an assembly size of 219 Mb with the N50 length of 542,540 bases. Using 2857 single nucleotide polymorphism (SNP) markers, this study investigated the genetic diversity and population structure of 80 *R. mucronata* accessions obtained from the mangrove forests in Thailand. The genetic diversity of *R. mucronata* was moderate (I = 0.573, Ho = 0.619, He = 0.391). Two subpopulations were observed and confirmed from both population structure and principal component analysis (PCA). Analysis of molecular variance (AMOVA) showed that there was more variation within populations than between them. Mean pairwise genetic differentiation (*F*_ST_ = 0.09) showed that there was not much genetic difference between populations. Intriguingly, the predominant clustering pattern in the *R. mucronata* population did not correspond to the Gulf of Thailand and the Andaman Sea, which are separated by the Malay Peninsula. Several factors could have influenced the *R. mucronata* genetic pattern, such as hybridization and anthropogenic factors. This research will provide important information for the future conservation and management of *R. mucronata* in Thailand.

## 1. Introduction

Mangroves are one of the most prolific marine ecosystems on Earth [1]. They perform vital biological tasks and provide numerous benefits to coastal areas. These forests, at the land–sea interface, serve as critical habitats for several marine and terrestrial species (including many valuable commercial species and juvenile reef fish) [2]. In addition, they provide vital ecological services such as coastal protection during storms, waste treatment by filtering terrestrial runoff, fisheries preservation, and carbon sequestration [3,4]. Globally, mangroves cover over 138,000 km^2^ of the coastal area and are found along tropical and subtropical coasts between roughly 25° N and 25° S. They are primarily found in Asia (38.7%), Africa (20%), and Latin America and the Caribbean (20.3%) [5]. Despite their importance as a physical protective barrier along the world’s coastlines, mangroves are also one of the marine ecosystems most vulnerable to global climate change. Human activity has been identified as the principal driver of global mangrove degradation in the twentieth century [6]. Forest clearing and exploitation for timber production and raw minerals, as well as increasing coastal population growth and urbanization, are all contributing factors [7]. Mangroves are regarded as one of the most endangered ecosystems [8,9]. Over the past four decades, it has been observed that between 20% and 35% of the world’s mangrove forests have vanished, and the rate of decline of mangrove areas is estimated to be approximately 1% per year [10]. As habitats disappear, wildlife populations become isolated and limited, which can result in the loss of genetic diversity by exacerbating genetic drift and decreasing gene flow. The decrease in genetic diversity will increase the probability of extinction for these populations [11,12]. Understanding the genetic diversity and population structure in mangrove communities is crucial for developing better methods for restoration and conservation [13].

*Rhizophora* is the most representative genus of the mangrove family *Rhizophoraceae*, and has a wide distribution over the Atlantic–East Pacific region, as well as the Indo-West Pacific region [14]. *R. mucronata*, also known as loop-root mangrove, red mangrove, or Asiatic mangrove, is a plant in Thailand’s mangrove forests that is extremely valuable to the community as a source of both wood and nonwood items [15], as well as the primary materials used for home construction and fuel wood [16]. Bark and leaf extracts of the *R. mucronata* plant have been used medicinally for centuries as an astringent, antiseptic, and hemostatic. This extract also possesses antibacterial, antiulcerogenic, and anti-inflammatory properties [17]. Although genetic and phylogeographic information on mangroves is increasing, the majority of studies have focused on populations of *Avicennia* and *Rhizophora* species in other Southeast Asian countries [14,18,19,20], with very few studies on mangrove communities along Thailand’s coast [21,22].

Over the past decade, the advances in sequencing technologies have provided genomic information to study population genetics in mangroves. A number of reference genomes of species in the family Rhizophoraceae have been reported [23,24,25,26,27,28,29]. Various molecular markers have been utilized to assess the genetic diversity of mangrove species, including amplification fragment length polymorphism (AFLP) [30], simple sequence repeats (SSR) [31,32] and randomly amplified polymorphic DNA (RADP) [33]. These approaches can be time-consuming, expensive, and produce a small number of markers. Single nucleotide polymorphisms (SNPs) are now the marker of choice due to their abundance, stability, and good distribution throughout the genome [34]. The restriction site-associated DNA (RAD-seq) is one of the reduced-representation sequencing approaches [35] to obtain a genome-wide, unbiased set of SNP markers by digesting target genomes with restriction enzymes. The assessment of genetic diversity in several plant species has been performed using the RAD-seq technique [36,37,38].

In this study, the whole genome of *R. mucronata* was sequenced and assembled as a reference genome. SNP markers that were generated from RAD-seq were utilized to determine the genetic diversity of *R. mucronata* individuals, as well as the population structure of these individuals along the coast of Thailand. This research we conducted will contribute to a better understanding of the genetic variability and current status of *R. mucronata* populations in Thailand.

## 2. Materials and Methods

### 2.1. Plant Materials

One *R. mucronata* was chosen as a representative species for reference genome sequencing in this work. The sample was taken from a natural mangrove forest in Ranong province (9°52′36.1″ N 98°36′11.5″ E) under the supervision of Thailand’s Department of Marine and Coastal Resources. The morphology of *R. mucronata* is presented in Figure 1.

For genetic diversity and population structure analysis, leaf samples from 80 *R. mucronata* trees were collected from the mangrove forests in 18 provinces along the Andaman and Gulf of Thailand coasts in Thailand: Phetchaburi (PBI; 3), Chanthaburi (CTI; 3), Chachoengsao (CCO; 5), Chumphon (CMP; 5), Nakhon Si Thammarat (NST; 4), Narathiwat (NWT; 2), Phang-nga (PNA; 6), Phuket (PKT; 5), Prachuap Khiri Khan (PKN; 2), Pattani (PTN; 4), Ranong (RNG; 5), Samut Sakhon (SKN; 2), Samut Songkhram (SKM; 6), Samut Prakan (SPK; 4), Satun (STN; 7), Surat Thani (SNI; 4), Trat (TRT; 8), and Songkhla (SKA; 5) (Appendix A). Prior to this study, we surveyed the geographical distribution of *R. mucronata* in Thailand. These collection sites were selected based on the presence of the geographical distribution of *R. mucronata* and their accessibility. The sample size varied among sites depending on the size of the populations in each site. All of the samples were collected between 2020 and 2021 and the collection sites were displayed in a geographic map using the QGIS software v3.24.2, as shown in Figure 2.

### 2.2. DNA Extraction

Fresh leaves of all *R. mucronata* accessions were used to extract genomic DNA by using the standard CTAB (Cetyl Trimethyl Ammonium Bromide) method, followed by cleaning up using a DNeasy Plant Mini Kit (Qiagen, Hilden, Germany). DNA concentration can be determined with the Qubit fluorometer (Thermo Fisher Scientific, Eugene, OR, USA) and a Qubit DSDNA BR Assay Kit (Invitrogen, Eugene, OR, USA). One *R. mucronata* accession was used to generate a reference genome sequence using the 10× Genomics technology with linked-read sequencing, a microfluidics-based technique, making it possible to extract long-range information from short-read sequencing data (10× Genomics; accessed 13 January 2023). The 10× Genomics library was constructed from approximately 1 ng of high-quality, high-molecular weight DNA, following the manufacturer’s instructions for the Chromium Genome Library Kit and Gel Bead Kit v2, the Chromium Genome Chip Kit v2, and the Chromium i7 Multiplex Kit (10× Genomics). The sequencing was performed using the Illumina HiSeq X Ten, generating paired-end reads at 150 bp.

### 2.3. RAD Library Preparation and Sequencing

To construct a RADseq library, approximately 1 ug of genomic DNA was used following the MGIEasy RAD Library Prep Kit Instruction Manual (MGI Tech, Shenzhen, China). Briefly, *Taq*I restriction enzyme was used to digest genomic DNA, then fragments were ligated with a unique barcoded adapter. The samples were pooled in an equimolar amount and subjected to PCR and quantification. Paired-end sequencing at 150 bp per read was performed on the MGISEQ-2000RS following the manufacturer’s protocol (MGI Tech).

### 2.4. Genome Assembly and Annotation

Following Illumina’s protocol for 150 bp paired-end sequences, the linked-reads DNA from 10× Genomics (10 Genomics, Pleasanton, CA, USA) was sequenced on an Illumina HiSeq X Ten (Illumina, San Diego, CA, USA). The whole genome assembly of *Rhizophora mucronata* was performed using SuperNova v2.1.1 with default settings (https://support.10xgenomics.com/de-novo-assembly/software/pipelines/latest/using/running (accessed on 5 January 2023)). Annotation was performed by following the pipeline reported in [39]. We identified the protein-coding sequence in the unmasked genome using EvidenceModeler (EVM) v1.1.1 [40] by combining ab initio prediction evidence, RNA-based prediction, and homology prediction. The RNA-based prediction approaches used RNA-seq evidence from *R. mucronata* under accession SRR5384658, and a transcriptome assembly was performed using PASA v2.4.1 [40]. The Analysis and Annotation Tool (AAT) [41] was utilized to align protein sequences from the public database to unmask the genome with the following species: rice (*Oryza sativa*), *Mimulus guttatus*, sesame (*Sesamum indicum*), black cottonwood (*Populus trichocarpa*), and flooded gum (*Eucalyptus grandis*). The ab initio prediction program Augustus v3.3.3 [42] was trained with *O. sativa*, *M. guttatus*, *S. indicum*, *P. trichocarpa*, and *E. grandis*. All gene prediction evidence was then combined by EvidenceModeler to generate consensus gene models using the following weights for each evidence type: PASA2—1, GMAP—0.5, AAT—0.3, and Augustus—0.3. To obtain high-confidence gene sets, we cross-checked the positions of annotated genes with those of known repeats, then we excluded any gene that shared more than 50% of its sequence with those repetitious in the list. Using OmicsBox v2.0.10 (https://www.biobam.com/download-omicsbox/ (accessed on 16 January 2023)), the remaining predicted genes were functionally annotated. UniProtKB/Swiss-Prot (swissprot v5) and the GenBank nonredundant database (nr v5) were chosen to align with the protein-coding sequence using local BLASTP, with an e-value threshold of 10^−5^. Gene ontology (GO) terms were extracted and assigned to *R. mucronata* sequences, while enzyme codes (EC) were extracted and mapped to Kyoto Encyclopedia of Genes and Genomes pathway annotations.

### 2.5. Repetitive Sequence Identification

RepeatModeler v2.0.1 (http://www.repeatmasker.org/RepeatModeler/ (accessed on 10 January 2023)) was used to construct a de novo repeat library for transposable element (TE) family identification in an unannotated genome assembly. RECON v1.08 and RepeatScout v1.0.5, which are both de novo repeat-finding programs, were used to identify the boundaries of repetitive elements and to build consensus models of interspersed repeats. To verify that repeat sequences in the library did not contain large families of protein-coding sequences that were not TEs, they were aligned to GenBank’s nr protein database using BLASTX, with an e-value threshold of 10^−6^. Repeat masking was performed on the assembled genome by RepeatMasker v4.0.6 (http://www.repeatmasker.org/ (accessed on 10 January 2023)) with the custom species-specific repeat library generated by RepeatModeler.

### 2.6. Genome Assembly Quality Assessment

We assessed the quality of genome assembly by aligning short-read DNA sequences and RNA sequencing reads (RNA-seq) to the assembly using minimap2 v2.17 [43] for DNA and HISAT2 v2.2.0 [44] for RNA-seq data. Genome completeness was assessed by comparing each annotated gene set to the orthologs in the Embryophyta OrthoDB release 10 using the benchmarking universal single-copy orthologs (BUSCO v4.0.5) [45,46].

### 2.7. Single Nucleotide Polymorphisms Identification

The RADseq data obtained from 80 *R. mucronata* accessions were used to identify single nucleotide polymorphisms (SNPs) following the Genome Analysis Toolkit (GATK) pipeline. Paired-end sequence reads from each accession were mapped to our genome assembly of *R. mucronata* (GenBank accession number JAJHUU000000000) using BWA v0.7.17-r1188 153 (https://github.com/lh3/bwa (accessed on 12 January 2023)) with default settings. GATK v4.1.4.1 with HaplotypeCaller mode was used for SNP calling. Finally, SNPs were filtered according to the following criteria: (a) a minor allele frequency greater than 0.1, (b) depth coverage ranging from 10× to 200×, and (c) less than 5% missing data. The remaining SNP markers were used to explore the genetic diversity and population structure of the 80 *R. mucronata* accessions.

### 2.8. Population Structure and Genetic Diversity

We employed STRUCTURE v2.3.4 with a Bayesian model-based approach [47] to analyze the population structure of *R. mucronata*. Twenty replicates were performed for each K value between 1 and 10, with a burn-in duration of 10,000 and a run length of 10,000 iterations. The number of subpopulations was determined by the ΔK method [48], which was implemented in the web-based STRUCTURE HARVESTER software [49]. Based on the optimal K value, CLUMPP v1.1.2 [50] was used to identify the average individual assignment probabilities for the 1000 replicates. The analysis of molecular variance (AMOVA) and population differentiation (*F*_ST_) was performed using ARLEQUIN v3.5 [51]. Principle component analysis was conducted to identify structure in the distribution of genetic variation within the population using the PCA function in R. The first two principal components were selected and plotted using R software v3.3.4 with the library ggplot2 [52]. The level of genetic diversity was assessed across the population of 80 *R. mucronata* defined by STRUCTURE, using GenAlEx v6.502 [53] to estimate the number of effective alleles (Ne), Shannon’s information index (I), observed heterozygosity (Ho), expected heterozygosity (He), and the inbreeding coefficient (*F*_IS_). Polymorphism information content (PIC) values for SNP markers were calculated using PowerMarker v3.25 [53].

## 3. Results

### 3.1. Genome Assembly and Genome Annotation

To generate a genome assembly of *R. mucronata*, one mature individual located in the protected mangrove forest in Ranong province was selected. High-quality genomic DNA was extracted and used for 10× Genomics linked-read library preparation. The sequencing of the genome produced 100.78 Gb from a total of 335,963,870 paired-end reads. The de novo assembly generated from the linked-read data was 219 Mb and contained 12,956 scaffolds. The longest scaffold was 4,666,015 bp and the N50 length was 542,540 bp (Table 1). The quality of the *R. mucronata* genome sequence was evaluated by aligning the short-read DNA sequences and RNA-seq data to the assembly. In total, 95.45% of the Illumina short-read sequences and 94.09% of the RNA-seq reads could be aligned to our genome assembly. In addition, the BUSCO3 pipeline was employed to assess the completeness of the conserved orthologs in the *R. mucronata* genome using the plant-specific database from Embryophyta OrthoDB release 10. Based on the BUSCO analysis, we discovered that our genome assembly retrieved 96.4% of the plant-specific orthologs, with 93.4% classed as complete and single-copy, 3.0% as complete and duplicated, 1.4% as fragmented, and 2.2% as missing. 

A combination of several types of evidence, including ab initio prediction, homology search, and transcript evidence from RNA-seq data, was integrated into the genome annotation pipeline. We obtained 28,500 predicted gene models in total, of which 25,308 were protein-coding genes. The average gene length in red mangrove is 2447 bases, with 4.64 exons (Table 2). We observed that coding regions were more GC-rich than that observed in introns and untranslated regions, 45.08% and 34.55%, respectively. The gene ontology (GO) had been assigned to 18,651 protein-coding genes (65.44%) (Appendix A). Protein phosphorylation was the most common gene ontology (GO) term associated with the biological process, followed by regulation of DNA-templated transcription and transmembrane transport, while integral component of membrane, nucleus, and cytoplasm were the most common terms associated with the cellular component. ATP binding, metal ion binding, and DNA binding were the three most important areas of molecular function (Appendix A). Of 25,308 predicted gene models, 68.25%, 29.48%, and 17.08% could be identified functionally using the Swissprot, EC, and KEGG databases, respectively (Appendix A).

### 3.2. Repeat Elements in the R. mucronata Genome

We determined that 27.85% (61.06 Mb) of the *R. mucronata* genome was composed of transposable elements by identifying and annotating repetitive sequences using RepeatModeler, RECON, and RepeatScout (Table 3). Of these repeated elements, 3.61 Mb (5.91%) were DNA transposons, 4.46 Mb (7.30%) were simple sequence repeats, 36.22 Mb (59.29%) were retrotransposons, and 16.77 Mb (27.5%) were unclassified repeat elements. Retrotransposons dominated the classified repetitive sequences, accounting for more than half of the total repeat elements in the genome. The majority of the retrotransposons (55.81%) were identified as long terminal repeats (LTR), which could be further subdivided into Copia-like (28.65%) and Gypsy-like (19.55%) elements (Table 3).

### 3.3. RAD-seq Data and SNP Identification

To generate SNP data, 80 *R. mucronata* accessions were sequenced and genotyped using RAD-seq. In total, we obtained approximately 755,820,058 raw reads, of which 710,562,039 (94.01%) were successfully mapped onto an assembled genome. The average number of reads generated per individual is 8,882,025, ranging from 958,901 to 95,664,876. Following the completion of the data preprocessing, the GATK pipeline uncovered an initial total of 1,065,643 SNPs. Following high-quality filtering, which enables no more than 5% missing data per locus, and minor allele frequency (MAF) more than 10%, a total of 2857 SNPs were retrieved for further analysis.

### 3.4. Population Genetic Structure

We utilized the Bayesian clustering method in conjunction with principal component analysis (PCA) from 2857 SNP markers in order to obtain an accurate picture of the *R. mucronata* population structure. The results of the STRUCTURE analysis revealed that the value of K = 2 had the highest ΔK (Figure 3A), which indicated that the population could be divided into two clusters. Each accession was placed in one of the two genetic clusters according to its membership probability (Q value), which needed to be greater than 0.6. The first cluster was represented by blue bars (Figure 3B) and contained 67 accessions that were collected from CCO, CMP, CTI, NST, NWT, PBI, PKN, PKT, PNA, PTN, RNG, SKA, SKM, SKN, SNI, SPK, STN, and TRT. The second cluster, represented by green bars (Figure 3C), contained the remaining 10 accessions and was composed of RNG, NST, SKM, and PNA. Three of the accessions were considered to be admixtures because their Q values were lower than 0.6, and they originated from SKM and SKN. Two distinct clusters of *R. mucronata* were found using principal component analysis, and these clusters accounted for 38% of the total variation (Figure 3C). The principal component analysis results were in agreement with the STRUCTURE analysis. 

### 3.5. Genetic Differentiation of Populations

In order to determine the genetic linkages between the STRUCTURE-based groups, we calculated pairwise *F*_ST_ values between clusters one and two. The *F*_ST_ values were moderate (*F*_ST_ = 0.09, *p* < 0.01), and the estimated gene flow among populations (Nm) was 2.528 on average. The comparatively high degree of gene flow corresponded to the relatively low level of genetic differentiation. These findings indicated that there was significant divergence within both clusters one and two. The AMOVA analysis was conducted based on two clusters that were inferred by the STRUCTURE method, and the results showed that the majority of the total variation occurred within populations and accounted for 90.90% of the total, whereas the variation between populations was only 9.10% of the total (Table 4). 

### 3.6. Genetic Diversity 

A total of 2857 SNPs were used to study the genetic diversity of *R. mucronata*. Genetic parameters, including polymorphism information content (PIC), gene diversity (GD), heterozygosity (Ho), and minor allele frequency, were calculated for the entire population (Figure 4A–D). The average PIC was 0.31, ranging from 0.16 to 0.38. The GD and Ho averaged 0.39 (0.18–0.5) and 0.61 (0.04–0.99), respectively. The average minor allele frequency was 0.31 and the majority value was 0.5. According to the STRUCTURE results, the *R. mucronata* populations were divided into two clusters. The genetic diversity of these clusters showed that the average values of Shannon’s information index (I), observed heterozygosity (Ho), and expected heterozygosity (He) were 0.573, 0.619, and 0.391, respectively. The number of effective alleles (Ne) averaged 1.701, ranging from 1.701 to 1.702. Cluster one had genetic diversity values (I = 0.578, Ho = 0.604, He = 0.394) similar to cluster two (I = 0.567, Ho = 0.633, He = 0.389). The average percentage of polymorphic loci was 99.09%, ranging from 98.19% in cluster two to 100% in cluster one. The inbreeding coefficient (*F*_IS_) values ranged from −0.538 (cluster one) to −0.451 (cluster two) with an average of −0.494 (Table 5), indicating the presence of an excess of heterozygotes.

## 4. Discussion

### 4.1. Genetic Structure and Genetic Differentiation

Understanding genetic diversity requires an understanding of population structure. In this study, we aim to evaluate the genetic structure of the mangrove *R. mucronata* in Thailand. We anticipated that geographical regions would divide this population’s structure into two distinct clusters. Intriguingly, the results suggested that the population could be divided into two clusters (K = two; Figure 3A), in accordance with the PCA analysis (Figure 3B), but the predominant clustering pattern in the *R. mucronata* population did not correspond to their geographical separation (the Gulf of Thailand and the Andaman Sea, which are separated by the Malay Peninsula). This is in contrast to previous studies on other mangrove species in Southeast Asia, which suggested that the Malay Peninsula served as a land barrier between populations occurring on its coasts, resulting in significant genetic differentiation. For instance, the genetic structure of *Bruguiera cylindrica* in Thailand was evaluated, and showed that their genetic structure corresponded to the Gulf of Thailand and the Andaman Sea according to the land barrier of the Malay Peninsula [22]. In addition, in a genetic study of *R. apiculata* using SNP markers, it was reported that the Gulf of Thailand population separated from the Andaman population [21]. Moreover, several studies have reported high differentiation between mangrove populations on both sides of the Malay Peninsula, including *Ceriops tagal* [54,55], *Bruguiera gymnorrhiza* [56,57], *Excoecaria agallocha* [58], *Sonneratia alba* [57,59], *Avicennia marina* [60], and *Avicennia alba* [57]. Meanwhile, our results showed that the Malay Peninsula has a minimal effect on the *R. mucronata* species. The results were consistent with the previously reported *R. mucronata* population structure in Southeast Asia, in which genetic differentiation was not found across the coasts of the Malay Peninsula [61]. Ng et al. [62] evaluated the levels and patterns of genetic variation of *R. mucronata* across the Malay Peninsula, and found that the Malay Peninsula is not a barrier to the gene flow of this species. In addition, Wee et al. [57] reported that significant genetic differentiation across the Malay Peninsula was detected in *A. alba*, *S. alba*, and *B. gymnorhiza*, but not in *R. mucronata*. In our study, we found that the *F*_ST_ values among the two clusters were not strong (*F*_ST_ = 0.09). According to Wright [63], the Nm was classified as high (≥1.0), medium (0.250–0.99), and low (0.0–0.249). Our results revealed that there was a high gene flow (Nm = 2.528) between the two clusters. Therefore, the high genetic flux among the populations led to their low or moderate genetic differentiation. The high level of gene flow indicated that physical surroundings, such as land, were not a barrier to gene flow. Several factors may have played a role in shaping the genetic pattern in *R. mucronata*. Possible factors could have contributed to the breakdown of the population structuring in *R. mucronata* across the Malay Peninsula, such as hybridization between isolated populations within a species that results in genetic admixture [64]. Hybridization may occur due to human activities that can lead to increased connectivity between populations [65]. Human activities affect genetic structure, either by altering the level of genetic diversity, or by causing large changes in specific allele frequencies [66]. In Thailand, the mangrove reforestation effort extensively planted *Rhizophora mucronata* and *Rhizophora apiculate*, since these species are widespread and predominate among mangrove trees. Furthermore, Rhizophora mangroves are hardy, rapidly developing, and produce viviparous propagules that are comparatively more viable than those of other mangrove species [10,67]. These two species were employed for mangrove transplantation and rehabilitation because their seeds are simple to plant in nurseries prior to transplantation [68]. These activities may have resulted in the acceleration of gene flow and genetic mixing between the populations of the Andaman Sea and the Gulf of Thailand coasts. Additionally, propagule dispersal is an important factor. Mangrove species have different propagule dispersal characteristics. Among all mangrove species, *R. mucronata* has the largest propagule with one of the longest flotation periods [61]. *R. mucronata* propagules can remain buoyant and viable in seawater for several months [69]. This could lead to potentially higher genetic connectivity across populations relative to species with shorter dispersal distances [69]. For instance, Wee et al. [57] tested the hypotheses that the genetic structure of four coastal mangrove species would reflect differences in dispersal potential across the Malay Peninsula. The four mangrove species are *Avicennia alba*, *Sonneratia alba*, *Bruguiera gymnorhiza*, and *R. mucronata*, which differ in propagule size and buoyancy. The results showed that significant east–west genetic differentiation across the peninsula was observed in *A. alba*, *S. alba*, and *B. gymnorhiza*, and the effect was most pronounced for the two species with lower dispersal potential (*A. alba*, *S. alba*). In contrast, the two species with higher dispersal potential (*B. gymnorhiza* and *R. mucronata*) exhibited much higher proportions of recent inter-population migration along the Malacca Strait. Thus, the high dispersal potential of propagules could be expected to result in high levels of population connectivity, and little genetic differentiation over large distances. Therefore, the combination of factors such as human activities, sea or ocean currents, and the dispersal ability of seeds may have influenced the genetic structure of *R. mucronata*, resulting in the *R. mucronata* population that did not correspond to their geographical separation.

### 4.2. Genetic Diversity

With the rapid loss of mangrove areas, understanding the genetic diversity of mangrove species is important for conservation and management. Several studies on mangrove species have reported genetic variation based mainly on SSR markers [14,70,71,72,73]. With the advancement of next generation sequencing technology, SNP markers have been increasingly applied for genetic study in plants [74,75]. In this study, we used 2857 SNP markers obtained by RAD-seq to evaluate the genetic diversity of the *R. mucronata* population. The PIC values of the SNP markers ranged from 0.16 to 0.38, with an average of 0.31. According to Botstein, et al. [76], the PIC can be divided into three categories: (1) when PIC > 0.5, the marker is considered to be highly polymorphic, (2) when 0.25 < PIC < 0.5, the marker is moderately informative, and (3) when PIC < 0.25, the marker is a low-information marker. Our markers were classified based on the PIC values as moderate or low-informative markers, which is similar to previous reported in a study on *B. parviflora* [39] and *R. apiculata* [21].

In this study, we assessed the levels and patterns of genetic variation of *R. mucronata* collected from mangrove forests in Thailand. Our results showed that the genetic diversity of *R. mucronata* was moderate (I = 0.573, Ho = 0.619, He = 0.391, Table 2). The same result has been found in previous mangrove studies, such as *Avicennia marina* worldwide [77], *A. marina* in the Malay Peninsula [60], and *Kandelia obovata* in China [78]. In addition, low genetic diversity of *Rhizophora* species was detected in several studies carried out in Japan, Brazil, Mexico, Southeast Asia, and the Indo-West Pacific region [14,31,32,61,70,79,80]. The level of genetic diversity of mangrove species is affected by several factors, such as the mating system, pollination, propagules dispersal, habitat fragmentation, climate change, and anthropogenic activities [79,81,82]. The mating system is one of the important factors influencing genetic diversity [83]. Outcrossing species have a much higher level of genetic diversity than self-pollinated species [84]. *R. mucronata* was a mixed-mating species and its mating system was predominantly outcrossing [14]. This could be an important factor that is often considered to influence the level of genetic diversity. However, the mating system is not the only factor that shapes genetic diversity. The genetic diversity of mangrove populations is influenced by intrinsic factors, such as biparental inbreeding, ongoing hybridization, and rate of gene flow through pollen and propagule dispersal, as well as by extrinsic factors, such as marine currents and tidal patterns [85]. Additionally, anthropogenic disturbances could be a possible factor that influences the level of genetic diversity of species [66]. The combination of this complex set of ecological features may shape the genetic diversity of *R. mucronata*.

## 5. Conclusions

In the present study, we investigated the genetic diversity and population structure of *R. mucronata* in Thailand’s coastal areas. The *R. mucronata* population has moderate genetic diversity (I = 0.573, Ho = 0.619, He = 0.391) and differentiation based on SNP markers, with the majority of variations occurring primarily within the population. The STRUCTURE and PCA analyses both indicate that *R. mucronata* populations can be divided into two genetically distinct groups. However, the results of two genetic groups did not correspond to the geographical separation between the Gulf of Thailand and the Andaman Sea. Several factors could have influenced the *R. mucronata* genetic pattern, such as hybridization and anthropogenic factors. This research will provide better information for the future conservation and management of *R. mucronata* in Thailand.

## Figures and Tables

**Figure 1 biology-12-00484-f001:**
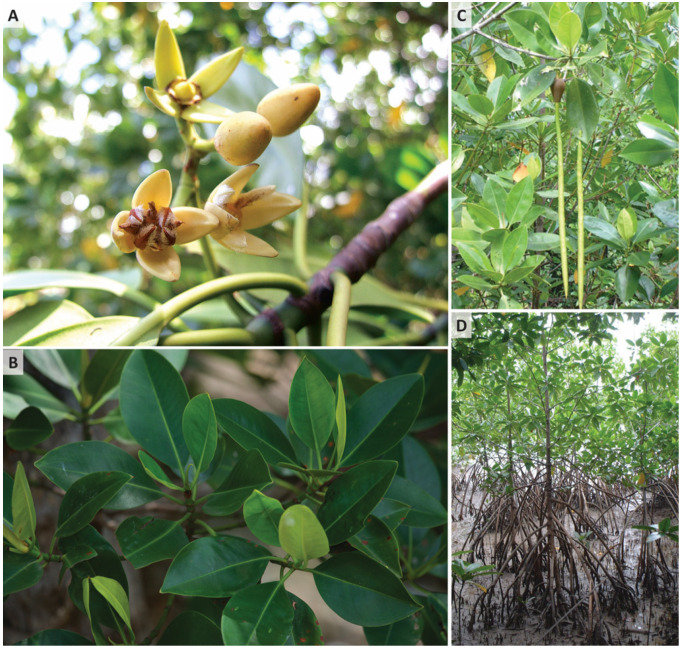
Morphology of *R. mucronata* in Thailand. (**A**) Flower. (**B**) Leaves. (**C**) Fruits. (**D**) Aerial root. Pictures taken by Wasitthee Kongkachana on 26 October 2020.

**Figure 2 biology-12-00484-f002:**
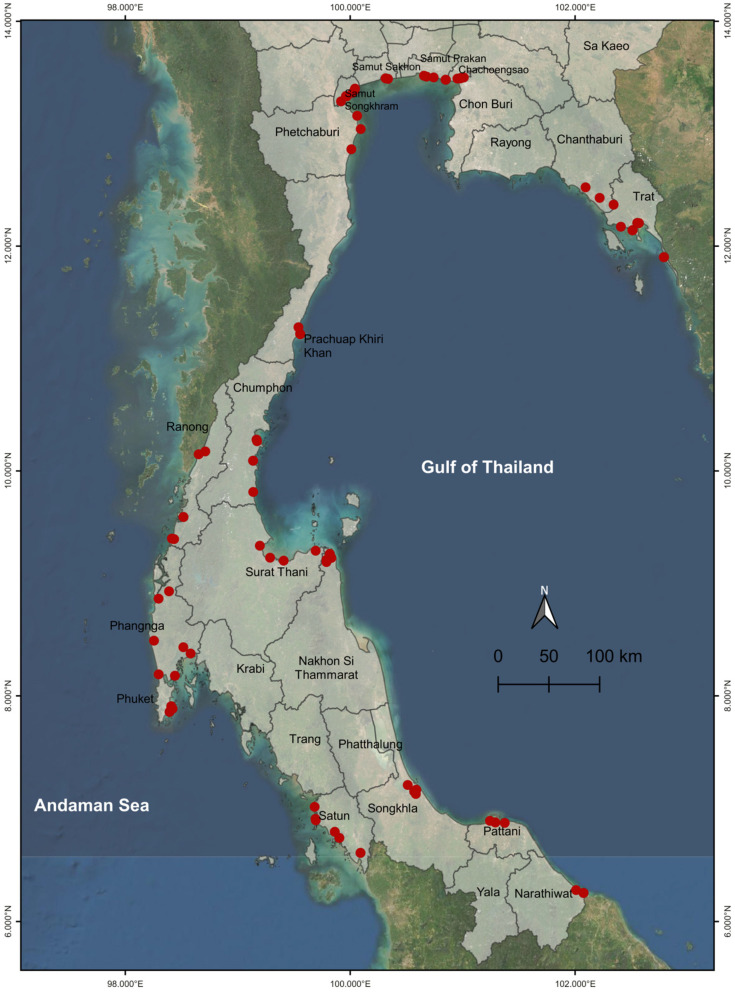
The geographical location of 80 *R. mucronata* accessions in Thailand. Collection sites in 18 provinces along the Andaman and Gulf of Thailand coasts in Thailand: Phetchaburi (PBI), Chanthaburi (CTI), Chachoengsao (CCO), Chumphon (CMP), Nakhon Si Thammarat (NST), Narathiwat (NWT), Phang-nga (PNA), Phuket (PKT), Prachuap Khiri Khan (PKN), Pattani (PTN), Ranong (RNG), Samut Sakhon (SKN), Samut Songkhram (SKM), Samut Prakan (SPK), Satun (STN), Surat Thani (SNI), Trat (TRT), and Songkhla (SKA). Red dots indicate collection sites at which the accessions were collected.

**Figure 3 biology-12-00484-f003:**
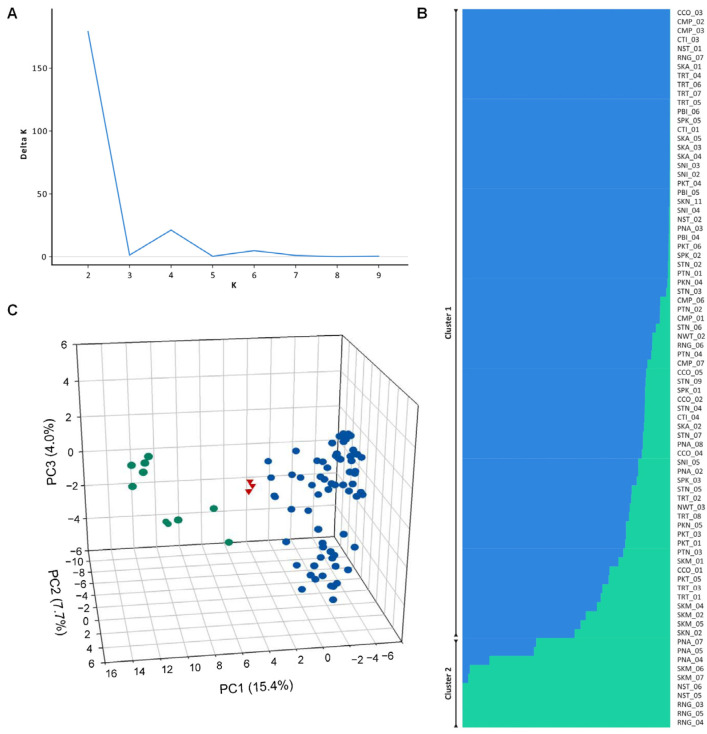
Population structure of 80 *R. mucronata* accessions using 2857 SNP markers inferred by STRUCTURE program. (**A**) Number of subpopulations indicated by the highest ΔK. (**B**) Proportion of clustering of individuals to two clusters. Each bar represents one accession. Blue and green colors within each bar indicate admixture among samples collected from the Andaman and Gulf of Thailand coast. (**C**) Distribution of 80 *R. mucronata* accessions on a scatter plot based on PC1, PC2, and PC3 from principal component analysis (PCA). The accessions are colored according to the two STRUCTURE clusters. The red triangles indicated the admixture accessions.

**Figure 4 biology-12-00484-f004:**
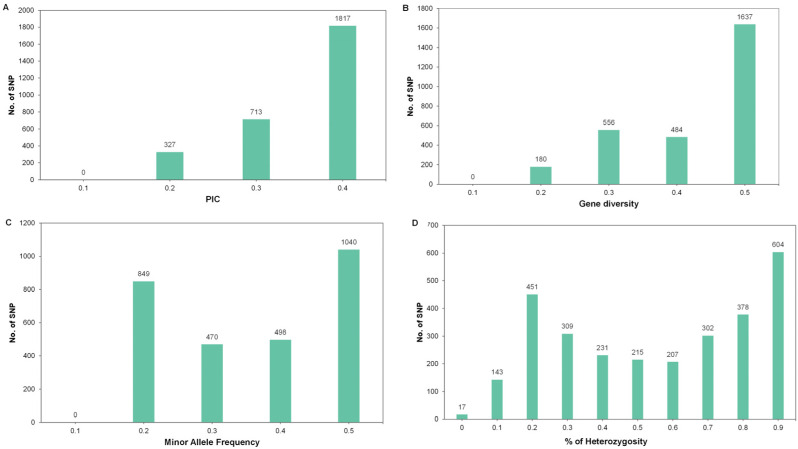
Distribution of (**A**) polymorphism information content (PIC), (**B**) genetic diversity, (**C**) minor allele frequency, and (**D**) percentage of heterozygosity for 2857 SNP markers among the 80 *R. mucronata* accessions.

**Table 1 biology-12-00484-t001:** *R. mucronata* genome assembly statistics.

	10× Genomics
N50 scaffold size (bases)	542,540
L50 scaffold number	108
N75 scaffold size (bases)	157,341
L75 scaffold number	287
N90 scaffold size (bases)	6738
L90 scaffold number	1569
Total (bases)	219,246,407
Number of scaffolds	12,956
Number of scaffolds ≥ 100 kb	340
Number of scaffolds ≥ 1 Mb	37
Number of scaffolds ≥ 10 Mb	-
Longest scaffold (bases)	4,666,015
GC content (%)	35
BUSCO evaluation (% completeness)	96.4

**Table 2 biology-12-00484-t002:** *R. mucronata* genome annotation statistics.

	*R. mucronata*
Number of predicted gene models	28,500
Total gene length (Mb)	69.73
Average gene size (nt)	2447
Average number of exons/gene	4.64
Total exon length (Mb)	31.21
Average exon length (nt)	235.60
GC content of exons (%)	45.08
Average number of introns/gene	3.64
Total intron length (Mb)	38.55
Average intron length (nt)	370.80
GC content of introns (%)	34.55

**Table 3 biology-12-00484-t003:** Repeat contents in the *R. mucronata* genome assembly.

Types of Repeats	Bases (Mb)	% of the Assembly	% of Total Repeats
DNA transposons	3.61	1.65	5.91
Retrotransposons:			
LINE	1.93	0.88	3.16
SINE	0.20	0.09	0.32
LTR: Copia	17.50	7.98	28.65
LTR: Gypsy	11.94	5.44	19.55
LTR: Others	4.65	2.12	7.61
Simple sequence repeats	4.46	2.03	7.30
Others	16.77	7.66	27.5
Total	61.06	27.85	

**Table 4 biology-12-00484-t004:** Analysis of molecular variance (AMOVA) of *R. mucronata* population.

Source of Variation	df	Sum of Squares	Variance Components	Percentage of Variation	*F*-Statistic
Among populations	1	2425.68	53.92	9.10	*F*_ST_ = 0.09 **
Within populations	158	85,072.54	538.43	90.90	
Total	159	87,498.22	592.35		

Notes. df: degree of freedom, *F*_ST_: genetic differentiation, ** statistical significance level at 99% (*p* < 0.01).

**Table 5 biology-12-00484-t005:** Genetic diversity measurement of two populations of *R. mucronata* accessions used in this study based on 2857 SNP markers.

Population	N	Ne	I	Ho	He	PPL	*F* _IS_
Cluster 1	70	1.701	0.578	0.604	0.394	100%	−0.451
Cluster 2	10	1.702	0.567	0.633	0.389	98.19%	−0.538
Overall	80	1.701	0.573	0.619	0.391	99.09%	−0.494

Notes. N: number of samples, Ne: number of effective alleles, I: Shannon’s information index, Ho: observed heterozygosity, He: expected heterozygosity, PPL: percentage of polymorphic loci, *F*_IS_: inbreeding coefficient.

## Data Availability

The genome sequence of *R. mucronata* was submitted to the National Center for Biotechnology Information (NCBI), the genome accession number was JAJHUU000000000.

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
