# Peer review of "Assessment of the Genetic Diversity and Population Structure of Rhizophora mucronata along Coastal Areas in Thailand"

_biology, 2023, doi:10.3390/biology12030484_

Round 1

Reviewer 1 Report

Although genetic diversity and population structure from mangrove plants have been studied since 2000s. This study is an important, well researched and well written article. The authors are to be complimented.

I have noted major revision to catch your attention.

1.       Please clarify why authors preferred to used SNP markers and PCA analysis than geographical samples to determine the populations or based on accessions only?

2.       Please explain why R. mucronata population did not correspond to their geographical separation, as other Rhizohporaceae did.

3.       Several factors could have influenced R. mucronata genetic pattern such as hybridization and anthropogenic factors did not sufficiently discuss in the ms, kindly provide deep discussion about this matter.

4.       Please discuss briefly SNP markers evaluate and possibility to adopt to other mangrove species.

Author Response

We thank the reviewer for your valuable comments on the manuscript and have edited the manuscript to address their concerns.

Point 1: Please clarify why authors preferred to used SNP markers and PCA analysis than geographical samples to determine the populations or based on accessions only?

Response 1: Because of their abundance, stability, and well-distribution throughout the genome, SNP markers are highly informative molecular markers that can identify genetic differences between and within populations. PCA (Principal Component Analysis) is a statistical technique for analyzing genetic variation and population organization. It enables the visualization of genetic links between individuals or populations, as well as the identification of patterns of genetic variation that may be related with various geographic or environmental factors. Although the understanding of genetic diversity and population structure using traditional approaches based solely on geographical samples or accessions it is not sufficient. Yet, they may not adequately represent the genetic variety within populations, which might be problematic. Due to variables such as gene flow, genetic drift, and natural selection, distinct populations may have similar or identical accessions.

Point 2: Please explain why R. mucronata population did not correspond to their geographical separation, as other Rhizohporaceae did.

Response 2: We have explained why the clustering pattern in the R. mucronata population did not correspond to their geographical separation in the discussion section in Line 398-415. The detailed are as follows: “Several factors may have played a role in shaping the genetic pattern in R. mucronata. Possible factors could have contributed to the breakdown of the population structuring in R. mucronata across the Malay Peninsula, such as hybridization between isolated populations within a species that results in genetic admixture [81]. Hybridization may occur due to human activities that can lead to increased connectivity between populations [82]. Human activities affect genetic structure either by altering the level of genetic diversity or by causing large changes in specific allele frequencies [83]. In Thailand, the mangrove reforestation project widely used the seedlings of R. mucronata in mangrove transplantation and rehabilitation. These activities may have resulted in the acceleration of gene flow and genetic mixing between the populations of the Andaman Sea and the Gulf of Thailand coasts. Additionally, propagule dispersal is one of the important factors because R. mucronata propagules can remain buoyant and viable for several months in seawater [84]. The propagules are likely to be dispersed across large distances by the prevailing ocean currents in this region [66]. Therefore, the combination of factors such as human activities, sea or ocean currents, and the dispersal ability of seeds may have influenced the genetic structure of R. mucronata, resulting in the R. mucronata population that did not correspond to their geographical separation.”

Point 3: Several factors could have influenced R. mucronata genetic pattern such as hybridization and anthropogenic factors did not sufficiently discuss in the ms, kindly provide deep discussion about this matter.

Response 3: We have provided more explanation about this matter in Lines 401–405. The detailed modifications are as follows: “Possible factors could have contributed to the breakdown of the population structuring in R. mucronata across the Malay Peninsula, such as hybridization between isolated populations within a species that results in genetic admixture [66]. Hybridization may occur due to human activities that can lead to increased connectivity between populations anthropogenic factors [67]. Human activities affect genetic structure either by altering the level of genetic diversity or by causing large changes in specific allele frequencies [68].”

Point 4: Please discuss briefly SNP markers evaluate and possibility to adopt to other mangrove species.

Response 4: SNP markers have been used to study the genetic diversity and population structure of a number of mangrove species, For example Avicennia marina, Rhizophora apiculata, and Bruguiera cylindrica. These studies have shown that SNP markers can provide a powerful tool for studying the genetic diversity and population structure of mangrove species, and for understanding their evolutionary history and adaptation to changing environments. However, in order to adopt SNP markers to other mangrove species, we have not yet conducted an analysis of those SNP marker sets, but we do intend to look into the feasibility of applying them to additional species of mangrove.

Reviewer 2 Report

This article provides the importance of the red mangrove tree (Rhizophora mucronata) in Thailand and highlights the study rationale and why this species should be examined. It is also substantial to investigate genetic diversity. However, it is likely to be required to provide deeper discussion and what are implications and further research direction or implications for future studies regarding red mangroves in Thailand, especially the discussion part is needed to be written results with sub-sections. I added some specific comments in the attached files. 

Author Response

We thank the reviewer for your valuable comments on the manuscript and have edited the manuscript to address their concerns.

Point 1: What is implication of summary and your research? is it the first research of genetic diversity regarding R. mucronata? or something others?

Response 1: Thank you for your comment. The implication of our research is that it sheds light on the genetic diversity of Rhizophora mucronate across the range of Thailand coasts, which is an important mangrove species with ecological and economic significance. Our study is not the first research on the genetic diversity of R. mucronata, but it contributes to the existing body of knowledge by using a novel set of genetic markers and analyzing populations from a wider geographic range than previous studies. Our findings suggest that there is more variation within populations than between them, which has important implications for the conservation and management of this species. Additionally, our results may be useful for future studies investigating the adaptive potential of R. mucronata in the face of changing environmental conditions.

Point 2: Globally, mangroves cover–

Response 2: We have edited it.

Point 3: all over the world? or many regions? it might be controversial in other regions such as New Zealand.

Response 3: According to “Three decades of global mangrove conservation- An overview” paper, mangroves still experience an annual loss of 0.2–0.7% between 2000 and 2012, and remain the most threatened ecosystem of the world.

Point 4: Can you please mention regarding image credit? taken by whom or sources of picture with dates.

Response 4: We have made mention regarding image credit.

Point 5: Can you please mention how many samples were collected by each site?

Response 5: We have added the sample size for each site in Line 111-115.

Point 6: add country name

Response 6: We have added country name.

Point 7: Can you please provide more information regarding 10× genomics sequencing?

Response 7: We have provided more information regarding 10× genomics sequencing in Line 137-145. The detailed modifications are as follows: “One R. mucronata accession was used to generate a reference genome sequence using the 10x Genomics technology with linked-read sequencing, a microfluidics-based technique, it is possible to extract long-range information from short-read sequencing data (10x Ge-nomics; accessed 13 January 2023). The 10x Genomics library was constructed from approximately 1 ng of high quality, high molecular weight DNA, following the manufacturer's instructions for the Chromium Genome Library Kit & Gel Bead Kit v2, the Chromium Genome Chip Kit v2, and the Chromium i7 Multiplex Kit (10× Genomics). The sequencing was performed using the Illumina HiSeq X Ten, generating paired-end reads at 150 bp.”

Point 8: Did you explain about AAT before?

Response 8: We have added the full name of AAT.

Point 9: Single Nucleotide Polymorphisms

Response 9: We have edited it.

Point 10: As a important part of this paper, can you please divide and write by sub-section, which is related to each result?

Response 10: We have divided the discussion section.

Point 11: Why? can you explain specific reason, evidence or relevant references supporting this opinion?

Response 11: We have added the relevant references to support this opinion and edited the sentence. The detailed modifications are as follows: “Possible factors could have contributed to the breakdown of the population structuring in R. mucronata across the Malay Peninsula, such as hybridization between isolated populations within a species that results in genetic admixture [66]. Hybridization may occur due to human activities that can lead to increased connectivity between populations [67]. Human activities affect genetic structure either by altering the level of genetic diversity or by causing large changes in specific allele frequencies [68].”

Point 12: Provide some numeric value regarding this (e.g., I = 0.573, Ho = 0.619, He = 0.391).

Response 12: We have edited it.

Point 13: better

Response 13: We have edited it.

Reviewer 3 Report

Line 106, please replace the picture of the leaves (B) with a more detailed, clear and high-resolution picture.

Figure 2, please give a good justification for the selected 18 provinces. Why only 18 provinces are selected? Does it represents the mangrove of Thailand? For example, maybe it has to do with the location of the selected provinces.

Line 337, please do the correction on the typo or misaligned sentences.

Table 2, delete the space between Line 260 and Line 261

Table 3, delete the space between Line 274 and Line 275

Table 4, delete the space between Line 326 and Line 327

Table 5, delete the space between Line 359 and Line 360

Line 395 to Line 397, please give more justification on this matter. On why the predominant clustering pattern in the R. mucronata population did not correspond to their geographical separation?

As for the discussion, more citations from other similar research and publications are needed to support the results. Preferably the latest research and publications dated from 2020 to 2023.

The English language used in the manuscript can be improved by using proofreading.

Author Response

We thank the reviewer for your valuable comments on the manuscript and have edited the manuscript to address their concerns.

Point 1: Line 106, please replace the picture of the leaves (B) with a more detailed, clear and high-resolution picture.

Response 1: We have revised Figure 1 (B).

Point 2: Figure 2, please give a good justification for the selected 18 provinces. Why only 18 provinces are selected? Does it represents the mangrove of Thailand? For example, maybe it has to do with the location of the selected provinces.

Response 2: We have provided an explanation of the collection sites in Line 116-119. The detailed modifications are as follows: “Prior to this study, we surveyed the geographical distribution of R. mucronata in Thailand. Therefore, these collection sites were selected based on the presence of the geographical distribution of R. mucronata and their accessibility. The sample size varied among sites depending on the size of the populations in each site.”

Point 3: Line 337, please do the correction on the typo or misaligned sentences.

Response 3: We have edited it.

Point 4: Table 2, delete the space between Line 260 and Line 261

Response 4: We have edited it.

Point 5: Table 3, delete the space between Line 274 and Line 275

Response 5: We have edited it.

Point 6: Table 4, delete the space between Line 326 and Line 327

Response 6: We have edited it.

Point 7: Table 5, delete the space between Line 359 and Line 360

Response 7: We have edited it.

Point 8: Line 395 to Line 397, please give more justification on this matter. On why the predominant clustering pattern in the R. mucronata population did not correspond to their geographical separation?

Response 8: We have explained why the clustering pattern in the R. mucronata population did not correspond to their geographical separation in the discussion section in Line 398-415. The detailed are as follows: “Several factors may have played a role in shaping the genetic pattern in R. mucronata. Possible factors could have contributed to the breakdown of the population structuring in R. mucronata across the Malay Peninsula, such as hybridization between isolated populations within a species that results in genetic admixture [81]. Hybridization may occur due to human activities that can lead to increased connectivity between populations [82]. Human activities affect genetic structure either by altering the level of genetic diversity or by causing large changes in specific allele frequencies [83]. In Thailand, the mangrove reforestation project widely used the seedlings of R. mucronata in mangrove transplantation and rehabilitation. These activities may have resulted in the acceleration of gene flow and genetic mixing between the populations of the Andaman Sea and the Gulf of Thailand coasts. Additionally, propagule dispersal is one of the important factors because R. mucronata propagules can remain buoyant and viable for several months in seawater [84]. The propagules are likely to be dispersed across large distances by the prevailing ocean currents in this region [66]. Therefore, the combination of factors such as human activities, sea or ocean currents, and the dispersal ability of seeds may have influenced the genetic structure of R. mucronata, resulting in the R. mucronata population that did not correspond to their geographical separation.”

Point 9: As for the discussion, more citations from other similar research and publications are needed to support the results. Preferably the latest research and publications dated from 2020 to 2023.

Response 9: We have added more citations from other similar research and publications in the discussion section. The publications that we have added are as follows:

Triest, L.; Satyanarayana, B.; Delange, O.; Sarker, K.K.; Sierens, T.; Dahdouh-Guebas, F. Barrier to Gene Flow of Grey Mangrove Avicennia marina Populations in the Malay Peninsula as Revealed From Nuclear Microsatellites and Chloroplast Haplotypes. Frontiers in Conservation Science 2021, 2, doi:10.3389/fcosc.2021.727819.

Wee, A.K.S.; Noreen, A.M.E.; Ono, J.; Takayama, K.; Kumar, P.P.; Tan, H.T.W.; Saleh, M.N.; Kajita, T.; Webb, E.L. Genetic structures across a biogeographical barrier reflect dispersal potential of four Southeast Asian mangrove plant species. Journal of Biogeography 2020, 47, 1258-1271, doi:https://doi.org/10.1111/jbi.13813.

Yang, Y.; Li, J.; Yang, S.; Li, X.; Fang, L.; Zhong, C.; Duke, N.C.; Zhou, R.; Shi, S. Effects of Pleistocene sea-level fluctuations on mangrove population dynamics: a lesson from Sonneratia alba. BMC Evolutionary Biology 2017, 17, 22, doi:10.1186/s12862-016-0849-z.

Shi, J.; Joshi, J.; Tielbörger, K.; Verhoeven, K.J.F.; Macel, M. Costs and benefits of admixture between foreign genotypes and local populations in the field. Ecology and Evolution 2018, 8, 3675-3684, doi:https://doi.org/10.1002/ece3.3946.

Round 2

Reviewer 1 Report

Overall, the authors have made substantial changes. I think this paper might proceed for publication.

Author Response

Thank you very much for your valuable comments. 

Reviewer 2 Report

I believe it is an important topic to provide genetic composition and variation of mangrove species for global and regional ecosystem values. 

I appreciate authors tried to improve the quality of the manuscript by following my several comments. 

There are several comments required to reflect the manuscript:

L 339 What is the fundamental reason? due to the most dominant mangrove species and/or most representative genus but threatened species?

L 402 Is it only a specific characteristic of R. mucronata's propagule dispersal? and is there any previous case study or research regarding differences of propagule dispersal of different types of mangrove species (e.g., propagule's size, shape, weight, or the Eddy covariance related to pollination or other environmental factors)? How did other mangrove species more affect genetic differentiation from case studies?

L 435 Still unclear what kinds of factors are likely to affect (or affected) the moderate genetic diversity of R. mucronata (e.g., hybridization and anthropogenic factors)? 

Author Response

Dear reviewer,

Thank you very much for your valuable comments. Below, we are addressing them in detail:

Point 1: L 399 What is the fundamental reason? due to the most dominant mangrove species and/or most representative genus but threatened species?

Response 1: We have provided the fundamental reason in Line 425-431. The detailed modifications are as follows: “In Thailand, the mangrove reforestation effort extensively planted Rhizophora mucronata and Rhizophora apiculata since these species are widespread and predominate among mangrove trees. Furthermore, Rhizophora mangroves are hardy, rapidly developing, and produce viviparous propagules that are comparatively more viable than those of other mangrove species [10,69]. These two species were employed for mangrove transplantation and rehabilitation because their seeds are simple to plant in nurseries prior to transplantation [70].”

Point 2: L 402 Is it only a specific characteristic of R. mucronata's propagule dispersal? and is there any previous case study or research regarding differences of propagule dispersal of different types of mangrove species (e.g., propagule's size, shape, weight, or the Eddy covariance related to pollination or other environmental factors)? How did other mangrove species more affect genetic differentiation from case studies?

Response 2: We have provided more explanation about this matter in Lines 435–451. The detailed modifications are as follows: “Additionally, propagule dispersal is one of the important factors. Mangrove species have different propagule dispersal characteristics. Among all mangrove species, R. mucronata has the largest propagule with one of the longest flotation periods [71]. R. mucronata propagules can remain buoyant and viable in seawater for several months [72]. This could lead to potentially higher genetic connectivity across populations relative to species with shorter dispersal distances [72]. For instance, [73] test the hypotheses that the genetic structure of four coastal mangrove species would reflect differences in dispersal potential across the Malay Peninsula. The four mangrove species are Avicennia alba, Sonneratia alba, Bruguiera gymnorhiza and R. mucronata, which differ in propagule size and buoyancy. The result showed that significant east–west genetic differentiation across the peninsula was observed in A. alba, S. alba and B. gymnorhiza, and the effect was most pronounced for the two species with lower dispersal potential (A. alba, S. alba). In contrast, the two species with higher dispersal potential (B. gymnorhiza and R. mucronata) exhibited much higher proportion of recent inter-population migration along the Malacca Strait. Thus, the high dispersal potential of propagules could be expected to result in high levels of population connectivity and little genetic differentiation over large distances [74].”

Point 3: L 435 Still unclear what kinds of factors are likely to affect (or affected) the moderate genetic diversity of R. mucronata (e.g., hybridization and anthropogenic factors)?

Response 3:  We have provided more explanation about this matter in Lines 485–493. The detailed modifications are as follows: “This could be one important factor that is often considered to influence the level of genetic diversity. However, the mating system is not the only factor that shapes genetic diversity. The genetic diversity of mangrove populations is influenced by intrinsic factors, such as biparental inbreeding, ongoing hybridization, and rate of gene flow through pollen and propagule dispersal, as well as by extrinsic factors, such as marine currents and tidal patterns [90]. Additionally, anthropogenic disturbances could be a possible factor that influences the level of genetic diversity of species [91]. The combination of this complex set of ecological features may shape the genetic diversity of R. mucronata.”